# Genome-Wide Transcription Start Sites Mapping in *Methylorubrum* Grown with Dichloromethane and Methanol

**DOI:** 10.3390/microorganisms10071301

**Published:** 2022-06-27

**Authors:** Bruno Maucourt, David Roche, Pauline Chaignaud, Stéphane Vuilleumier, Françoise Bringel

**Affiliations:** 1Génétique Moléculaire, Génomique, Microbiologie, Université de Strasbourg, UMR 7156 CNRS, 67000 Strasbourg, France; bruno.maucourt@alumni.unistra.fr (B.M.); p.chaignaud@alumni.unistra.fr (P.C.); vuilleumier@unistra.fr (S.V.); 2LABGeM, Génomique Métabolique, Génoscope, Institut de Biologie François Jacob, Commissariat à l’Energie Atomique (CEA), CNRS, Université Evry, Université Paris-Saclay, 91057 Evry, France; droche@genoscope.cns.fr

**Keywords:** methylotrophy, dehalogenation, organohalide pollutant, dichloromethane, transcriptional start site, dRNA-seq, methanol, genome, gene expression

## Abstract

Dichloromethane (DCM, methylene chloride) is a toxic halogenated volatile organic compound massively used for industrial applications, and consequently often detected in the environment as a major pollutant. DCM biotransformation suggests a sustainable decontamination strategy of polluted sites. Among methylotrophic bacteria able to use DCM as a sole source of carbon and energy for growth, *Methylorubrum extorquens* DM4 is a longstanding reference strain. Here, the primary 5′-ends of transcripts were obtained using a differential RNA-seq (dRNA-seq) approach to provide the first transcription start site (TSS) genome-wide landscape of a methylotroph using DCM or methanol. In total, 7231 putative TSSs were annotated and classified with respect to their localization to coding sequences (CDSs). TSSs on the opposite strand of CDS (antisense TSS) account for 31% of all identified TSSs. One-third of the detected TSSs were located at a distance to the start codon inferior to 250 nt (average of 84 nt) with 7% of leaderless mRNA. Taken together, the global TSS map for bacterial growth using DCM or methanol will facilitate future studies in which transcriptional regulation is crucial, and efficient DCM removal at polluted sites is limited by regulatory processes.

## 1. Introduction

DCM (CH_2_Cl_2_) is a chlorine-containing one-carbon compound massively produced as a solvent feedstock for industrial synthesis of other chemicals, and for other applications such as varnish and paint removal [1]. According to the classification and labelling of chemicals (GHS), DCM is possibly carcinogenic to humans [1]. DCM also contributes to ozone layer depletion, like other human-produced chlorocarbons [2]. Nonetheless, DCM has not been included in the Montreal protocol of banned ozone-depleting gases due to its short atmospheric lifetime (around 6 months).

DCM-degrading strains have been isolated from industrial sites to drive biological environmentally friendly processes for DCM removal at contaminated sites [3,4]. DCM degraders such as the methylotrophic Alphaproteobacterial strain *Methylorubrum extorquens* DM4 achieve mineralization of DCM upon growth, with this compound as the sole source of carbon and energy [5,6]. Strain DM4 is a pink-pigmented facultative methylotroph able to also use several other C1 compounds, such as the reference methylotrophic growth substrate methanol, as well as multi-carbon compounds (e.g., succinate, acetate). Strain DM4 is suitable for genetic analysis (e.g., site-directed allelic gene replacement and replication of plasmids for classical molecular biology studies) [7], and has thus become the model reference strain for investigations of bacterial DCM degradation [8] and the identification of genes essential for growth with DCM. First and foremost, *M. extorquens* DM4 features the *dcmA*-encoded dehalogenase DcmA, whose product, formaldehyde, enters into central methylotrophic metabolism. Gene *dcmA* is highly conserved among DCM-degrading strains [9] and part of the 5.6 kb-long *dcm* gene cluster flanked by insertion elements [10]. The genome of strain DM4 consists of a 5.94 Mb chromosome and two plasmids: p1METDI and p2METDI of 141.5 and 38.6 kb, respectively [6]. On average, CDS length is 887 bp and intergenic sequences 183 bp long, with an overall gene density of 83.3%. Average GC content is 68.0%, for a total of 6035 genes. Expressed proteins and their termini were previously identified using a differential proteomics approach combined with a doublet N-terminal-oriented proteomics (dN-TOP) strategy. This allowed the identification of 47% of the total predicted proteins with experimental confirmation of 259 hypothetical proteins, discovery of 39 new proteins, and correction of 78 erroneous predicted N-termini [11].

Strain DM4 and other bacteria that dehalogenate organohalides such as DCM have to cope with stresses associated with dehalogenation activity (intracellular increase in halides (e.g., Cl, Br, I) and acid production), organohalide toxicity (solvent stress), as well as changes in metabolic flux. Collectively, these stresses have a strong impact on gene regulation and the activity of regulatory factors, with extensive genome-wide modulation of gene expression [12,13].

Modulation of transcription initiation is the first step in the regulation of gene expression. It is crucial for genes that are essential for growth and global regulation of metabolism, and thus for bacterial responses to stress and environmental variations. In the case of growth with DCM, for instance, *dcmA* gene expression is accompanied by high transcription levels of *clcA* encoding chloride/proton antiporter *clcA*, involved in dealing with chloride stress associated with DcmA dechlorination activity [14]. In addition, a study of gene expression profiles by RNA-seq at a high sequencing depth of strain DM4 grown with DCM, compared to growth with methanol, revealed 190 genes with higher transcript abundance with DCM [12]. However, promoter regions have not yet been extensively investigated in the *Methylorubrum* genus so far, and only very few transcriptional start sites (TSSs) have been experimentally determined, including in DCM-degrading strains [15,16,17,18,19,20,21,22,23].

Prediction of promoter regions based solely on computational methods does not allow full identification of transcription features of interest, such as alternate promoters and promoters that govern antisense transcription or are located within genes. Such features, however, are highly relevant for gene expression and transcription/translation coupling in prokaryotes (for a review see [24]). Experimental identification of TSSs, in contrast, greatly facilitates prediction of the corresponding promoters, discovery of associated regulatory elements, and precise delimitation of 5′untranslated region (5′UTR) length. Recently, differential RNA-seq (dRNA-seq) was developed as a high-throughput sequencing strategy to simultaneously map all the TSSs in the transcriptome, without a priori and at the scale of the genome (for a review see [25]). Briefly, this method differentiates primary transcript with 5′-triphosphate ends from processed transcripts that carry 5′-monophosphate or 5′-hydroxyl ends. Genome-wide TSS mapping was successfully used to characterise global gene expression in a wide variety of bacteria [25], but only a restricted number of methylotrophic species, including Firmicutes (*Bacillus methanolicus* MGA3 [26]; *Bacillus amyloliquefaciens* strains XH7 [27] and FZB42 [28]), Proteobacteria (*Bradyrhizobium japonicum* USDA 110 [29]), and Actinobacteria (*Mycobacterium smegmatis* [30]). Moreover, and to the best of our knowledge, no TSS genome-wide mapping of a bacterium capable of organohalide degradation has yet been published. In this study, dRNA-seq was used to map the TSS of the genome of *M. extorquens* DM4 from methylotrophic mid-exponential phase DCM and methanol cultures of the strain.

## 2. Materials and Methods

### 2.1. Cell Growth and RNA Extraction

*M. extorquens* wild-type strain DM4 was grown aerobically at 30 °C in M3 medium supplemented with methanol or DCM provided at 10 mM as the sole source of carbon and energy, in independent duplicates of 220 mL in 1.2 L Erlenmeyer flasks closed with gas-tight screw caps with Supelco Mininert R valves (Thermo Fisher Scientific, Illkirch, France), as described previously [31]. Cells were harvested at mid-phase (OD at 600 nm of 0.15), and RNA was extracted and treated with a TURBO DNA-free kit (Thermo Fisher Scientific, Illkirch, France) to remove residual genomic DNA. Ribosomal RNAs were depleted using the Ribo-Zero™ Magnetic Kit for Bacteria (Epicentre, Madison, WI, USA). Integrity of RNAs was validated after high-resolution automated electrophoresis of the RNA samples using the Agilent 2100 Bioanalyzer system. Two biological replicates were performed for each of the two tested conditions.

### 2.2. Construction of 5′-End-Mapping Libraries and cDNA Sequencing

Vertis Biotechnologie AG (Germany) constructed the libraries in a strand-specific manner, as described previously [32]. Briefly, rRNA-depleted RNA samples were poly(A)-tailed using poly(A) polymerase. After treatment with 5′ monophosphate-dependent RNA exonuclease (TEX, Euromedex, Souffelweyersheim, France), the four samples were split in two parts: one was treated with RNA 5′ polyphosphatase (5′PP) to remove the 5′PPP structures while the other was left untreated and used for “+” and “−” library construction, respectively (Appendix A). The RNA adapter was ligated to RNA with a 5′ monophosphate end. Oligo(dT)-adapter primer was used to perform synthesis of first strand cDNA with M-MLV reverse transcriptase. Resulting cDNAs were PCR-amplified with high-fidelity DNA polymerase and barcoded using 3′ sequencing adaptors. After cDNA purification using the Beckman Coulter Agencourt AMPure XP kit (Thermo Fisher Scientific, Illkirch, France) and quality checking using capillary electrophoresis (Shimadzu MultiNA microchip electrophoresis system), cDNA was pooled and sequenced on an Illumina NextSeq 500 system using 50 bp read length.

### 2.3. Identification of 5-Ends and Discrimination between Transcription Start Sites (TSS) and Cleavage Sites

Sequence raw data (.fastq) were quality-checked using FastQC (version 0.11.5, Babraham Institute, Cambridge, UK) and aligned to the reference genome with the Burrows-Wheeler Alignment tool (Version 0.7.4) [33]. A majority of the obtained reads were discarded, as they did not match the genome and corresponded to chimerical polyA sequences generated during library construction. Only aligned reads were used for TSS assignment using TSSAR (http://rna.tbi.univie.ac.at/TSSAR; accessed on October 2018) [34]. For each culture condition and genomic position on one of the three replicons (chromosome 5.94 Mb, plasmids p1METDI 142 kb and p1METDI 39 kb) present in genome of *M. extorquens* DM4, TSSAR assigned a TSS only if a significant difference between the two banks was observed based on the Skellam distribution. When the most stringent TSSAR parameters were used, i.e., 10 reads (noise for the minimal number of reads) and 1 nucleotide (merge for the maximal distance between two adjacent TSSs to be fused into a single TSS), a number of assigned TSSs equivalent to the number of predicted *M. extorquens* DM4 annotated genes (5988) [6] were obtained at a *p*-value of 0.01. Visualisation of the genome-wide nucleotide localisation of assigned TSS on the reference genome was performed with IGV (Integrative Genomics Viewer) [35]. When, for the same gene, several TSSs distant by less than 10 nt were observed with similar read numbers, manual validation of the TSSAR-detected TSS was performed. In such cases, the TSS with the most reads above noise at its 5′-end was validated, in particular when also found in the other culture condition, a criterion not taken into account by TSSAR. To retrieve genome-wide assigned TSS sequences upstream of the promoter region and the 5′UTR, two R packages were used, GenomicRanges [36] and seqinR [37]. Transcription start sites were categorised according to TSSAR as primary TSS (or gene TSS), antisense TSS, internal TSS and orphan TSS, according to their locations relative to annotated genes [34].

### 2.4. Promoter Motif Discovery

Identification of −10 and −35 regions was performed by scanning upstream sequences of the chromosomal 2261 predicted P and IP TSS for conserved motifs using MEME Differential Enrichment mode [38]. For the −10 region, the 20 nt sequences immediately upstream of predicted TSSs were used as the positive set, and the 20 nt downstream sequences of the same length at position +100 were used as the negative set. For the −35 region, the 20 nt sequences at −15 nt upstream of predicted TSS were used as the positive set, and the 20 nt downstream sequences of the same length at position +100 were used as the negative set. For each region, we assumed that each sequence in the dataset contains exactly one occurrence of each pattern using MEME’s “One occurrence per sequence (oops)” option [38].

### 2.5. Analysis of 5′Untranslated Regions (5′UTR)

The 5′-UTR region was defined as the sequence between the TSS and the start codon. 5′-UTR generally contains RBS, defined as the sequence that is partially complementary to 16S rRNA guiding the translation machinery to align with the start codon of the open reading frame (ORF) [39]. Leaderless mRNA represent mRNA with 5′-UTR of 0 to 9 nt in length, suggesting that transcription starts nearby the start codon of the corresponding ORF.

### 2.6. Data Deposition

RNA-seq data were deposited in the ArrayExpress database at EMBL-EBI (www.ebi.ac.uk/arrayexpress, accessed on 21 June 2022) under accession number E-MTAB-11726. Overview of mapped reads, and coverage directly plotted on IGV genome browser are available on the MicroScope platform (https://mage.genoscope.cns.fr/microscope/mage/index.php?), Transcriptomics tab, RNAseq project TSS_VXTYE3_Mextorquens_DM4.

## 3. Results and Discussion

### 3.1. Mapping and Annotation

Around 17 million reads were aligned to the genome of strain DM4, with similar numbers for each of the four replicates (Table 1). The unmapped reads were mostly poly(A) sequences that resulted from unspecific poly(A) tailing during library construction (Appendix A, step C). TSSs manually filtered for false positives (1597) and TSSs located upstream of tRNA (74) were removed (Appendix A, sheet called ‘removed TSS’). A total of 7231 TSSs were found evenly distributed along *M. extorquens* DM4 genome composed of a circular chromosome and two plasmids (Figure 1). Similar numbers of detected TSS were found for other bacteria [40], including Proteobacteria with a similar-sized genome [41].

TSSs were categorised into gene TSS (also called primary TSS), antisense TSS, internal TSS, and orphan TSS with a distribution shown in Figure 2. Gene TSSs account for the majority of the detected TSS located nearby annotated genes. Putative orphan TSSs represented 17% of detected TSSs (Figure 2). Orphan TSSs may result from TSS located further from annotated genes than the arbitrary 250 nt cut-off length applied in this study, from undetected and non-coding genes, or associated with antisense transcription [32]. TSSs on the opposite strand of CDS account for 31% of all identified TSS. Similarly, antisense TSSs represent 37% of all TSSs detected in *Escherichia coli* using the dRNA-seq approach [42]. This suggests that antisense transcription, also referred to as permissive transcription, may occur massively in *M. extorquens* DM4, a process that is widespread among Prokaryotes. So far, only the function of a few asRNA and sense/antisense overlapping genes coding for proteins have been identified [43,44].

Upstream of the initiation codon, canonical bacterial mRNA contains 5′UTR involved in transcript stability and translation efficiency. Analysis of the distance of gene TSS to the start codon revealed an average of 84 nt with a median value of 64 nt in *M. extorquens* DM4 (Figure 3). A similar mean length of 5′UTR was found in other bacteria, including *Sinorhizobium meliloti* [45]. Prokaryotes also produce leaderless mRNA that lack 5′UTR but are nevertheless competent for translation, with overlapping transcription and translation start sites. In the literature, the proportion of leaderless mRNA varies from around 0.5% in *B. methanolicus* MGA3 [26] and *E. coli* BW25113, up to 47% in *Deinococcus desertii* RD19 [46]. In this study, 7% of *M. extorquens* mRNA were leaderless (Figure 3). This percentage may represent an underestimation, as leaderless RNA, when resulting from cleavage of the 5′UTR, would not have been detected (i.e., processed mRNA was excluded from library “+”, see Appendix A). 

Conserved promoter motifs were searched for upstream of the 2261 predicted P and IP TSS of the chromosome, using MEME in Differential Enrichment mode. The −35 element consensus sequence found in *M. extorquens* resembles that of canonical *E. coli* (Figure 4). On the other hand, compared to the *E. coli* canonical −10 element, the highly conserved −7 thymidine nucleotide is absent in *M. extorquens*, as also found in other Alphaproteobacteria *Zymonmonas mobilis* and *Caulobacter crescentus* [47,48].

### 3.2. Focus on TSS and Promoter Region Associated with Genes of One-Carbon Metabolism

Before this study, only a few TSSs had been identified in *Methylorubrum/Methylbacterium* strains. For dichloromethane degradation, in the intergenic region of the divergently transcribed *dcmR*-*dcmA* genes that encode the dichloromethane dehalogenase DcmA and its transcriptional regulator DcmR, three TSSs were previously identified using nuclease S1 mapping. A unique TSS was detected for the promoter of *dcmA* (P_A_). Two TSSs were found for gene *dcmR* corresponding to promoters P_R1_ and P_R2_ [16]. In this study, using dRNA-seq, three TSSs (TSSAR class P) were detected corresponding to P_A_, P_R1_ and P_R2_ (Table 2). The TSS localisation for P_R2_ was identical. A 2–4 nt difference was observed for P_A_ and P_R1_, respectively (Appendix A). These differences may result from the use of RNA generated from different DNA templates (plasmid [16] versus native genomic localization in this study). When tested in a dichloromethane degrader affiliated to another genus, *Methylophilus* sp. strain DM11, primer extension on total RNA also identified a unique P_A_ promoter region. Nonetheless, a 53 nt shorter 5′UTR was predicted for P_A_ promoter in strain DM11 compared to strain DM4 [23] (Table 2).

Table 2 compares the identified TSSs detected in this study with *M. extorquens* DM4 to those characterized earlier in other strains (*M. extorquens* AM1; *Methylobacterium organophilum* strain XX) using nuclease S1 mapping, run-off or primer extension. In some cases, the correlation is excellent. This was the case for genes involved in methanol utilisation of the *mxa* operon as found for P*_maxF_* in *M. extorquens* AM1 [15,20] (Table 2). Primer extension applied to *M. organophilum* also identified P*_maxF_* despite sequence variation downstream of the −10 box and a 2 nt longer predicted 5′UTR [21] compared to other strains (Table 2).

Primer extension of the 5′end of *M. extorquens* AM1 *qsc2* (*mtkA-mtkB-ppc-mcl*) operon identified two TSSs associated with P*qsc2-1* and P*qsc2-2* promoters, predominant in RNA extracted from cultures in methanol and succinate, respectively [19]. In this study, only one TSS was detected 31 nt upstream of the start codon of *mtkA*. It most likely corresponds to the TSS of P*qsc2-1*, resulting in a 46 nt-long 5′UTR. When the sequence corresponding to P*qsc2-2* was searched in the genome of strain AM1 [6], a stretch of 30 nt separated the −35 and −10 boxes. The length of this spacer sequence is atypical of the bacterial consensus spacer region that generally includes 17 nt with variation between 15 to 20 nt [47]. This suggests a loss of P*qsc2-2* activity in strain AM1, and its minor role for methylotrophic growth conditions, as suggested previously [19].

Further, a newly identified TSS (TSSAR class P) was detected upstream of a key gene in C1 assimilation, gene *sga* encoding serine-glyoxalate aminotransferase, in addition to the previously detected TSS [19] (Table 2, P*qsc1* orphan-type TSS in this study). In two other cases, previously detected TSSs could not be confirmed here as class P TSS using dRNA-seq. Instead, TSSs were detected in upstream genes, suggesting co-transcription with adjacent genes (Table 2, see *orf181* and *pqqD*). Co-transcription of genes *ileS* and *lspA* (referred as *orf181* in [17]) is indeed likely, since these genes overlap in both *M. extorquens* DM4 and AM1 genomes, and RT-qPCR targeting this co-transcription had not been tested [18].

Altogether, the prediction of different promoter regions based on dRNA-seq correlated with those previously identified using different methods, in most cases. Nonetheless, we found some differences, which may originate from experimental design (e.g., cloned plasmid-borne promoters compared to native genomic location [49]), sequence variations between different strains, erroneous identification, and tested growth conditions with alternative TSSs. 

Besides the GATC(C/G)ATAGCCT motif present in the *dcmA* and *dcmR* promoters (P_A_ and P_R1_), no regulatory sequence associated with dichloromethane utilisation has been proposed so far [16]. Of the genes found with differential transcript abundance in cultures of strain DM4 grown with dichloromethane compared to growth with methanol [12], some may be co-regulated at the transcription level and harbour shared DNA motifs within their promoter regions and 5′UTR sequences. Among 69 genes with higher mRNA abundance upon growth with DCM compared to with methanol, a total of 33 TSSs were detected, whereas 16 TSSs were identified for 118 genes more expressed on methanol (Appendix A). However, no new remarkable DNA motifs were detected within 5′UTR of putatively regulated expressed genes using MEME.

## 4. Conclusions

The detailed TSS of DCM-degrading strain *M. extorquens* DM4 map obtained here establishes a valuable complement to genome sequence and proteome data for the strain, with different nucleic acid motifs associated with gene expression, such as consensus sequences found in promoter regions, mapped out in unprecedented detail (Figure 4). More generally, it provides a broad genome-level foundation for future in-depth investigations in the genus *Methylorubrum*, not only for gene expression and organisation but also for gene discovery. This rich dataset will facilitate future studies of genes of interest as well as systems biology investigations of methylotrophic metabolism, and in particular on the utilisation of chlorinated C1 compounds and corresponding adaptative mechanisms.

## Figures and Tables

**Figure 1 microorganisms-10-01301-f001:**
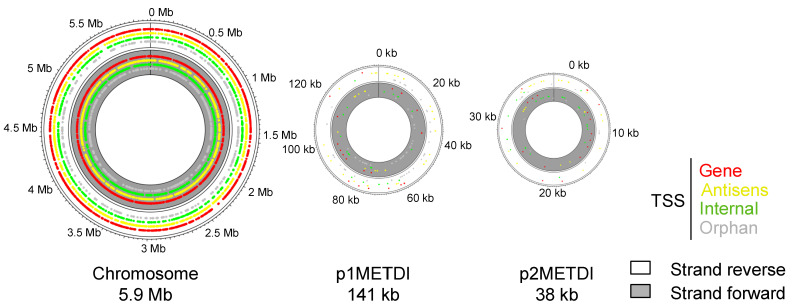
Distribution of the experimentally detected transcription start sites on the three replicons of *M. extorquens* DM4.

**Figure 2 microorganisms-10-01301-f002:**
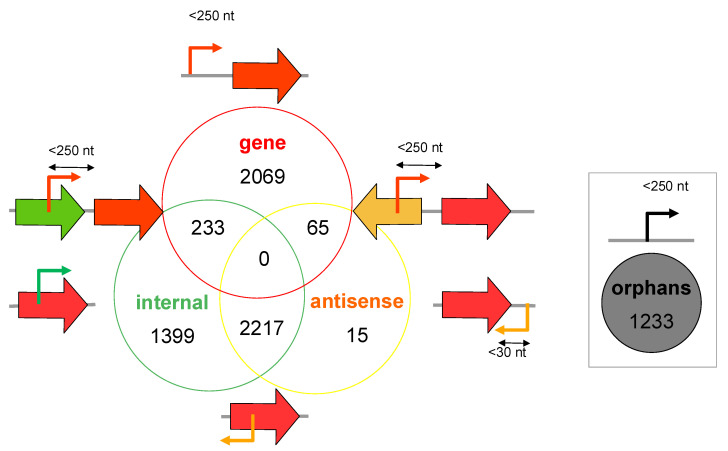
Genome-wide experimentally detected transcription start sites in *M. extorquens* DM4. TSS categories were defined using TSSAR [34]. Primary TSSs, also called gene TSSs (red circle), locate within 250 nucleotides upstream of an annotated gene. Internal TSSs (green circle) locate within a gene on the sense strand. Antisense TSSs (yellow circle) locate within a gene, or at less than 30 nucleotide distance of a gene, on the antisense strand. This class further splits into Ai and Ad, for internal antisense and downstream antisense, respectively. Orphan TSSs (grey circle) do not locate nearby annotated genes.

**Figure 3 microorganisms-10-01301-f003:**
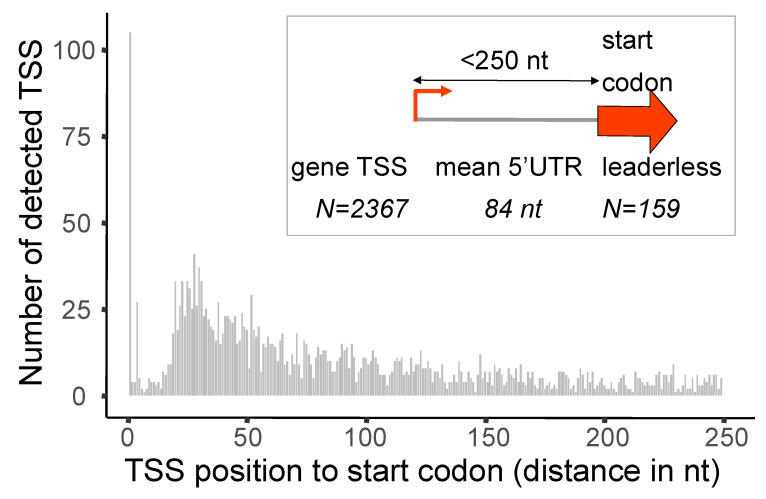
Length distribution of the 5′ untranslated region (5′UTR). A 0 distance to start codon represents TSS of leaderless transcripts. In this study, the maximum 5′UTR length was 250 nt.

**Figure 4 microorganisms-10-01301-f004:**
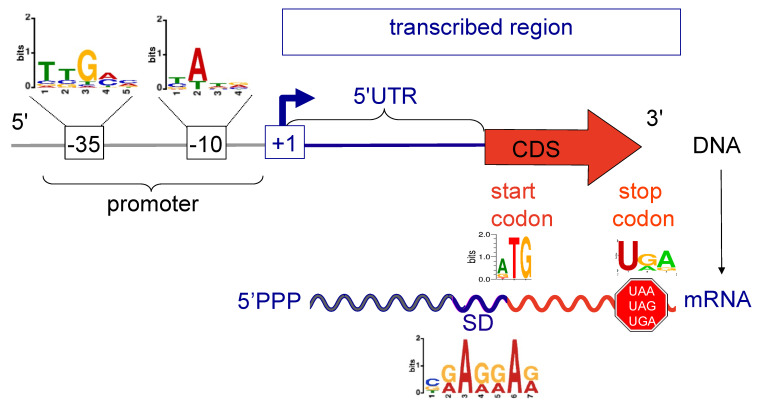
Remarkable nucleic acid motifs associated with *M. extorquens* DM4 gene expression. Transcription features (this work) and translation features [11] are shown. The −35 and −10 consensus sequences were deduced from TSS (P and IP) on the chromosome.

**Table 1 microorganisms-10-01301-t001:** Number of aligned reads on *M. extorquens* DM4 genome.

Bank	Methanol		Dichloromethane	
	Replicate 1	Replicate 2	Replicate 1	Replicate 2
“+”	0.6 × 10^6^	0.3 × 10^6^	0.8 × 10^6^	0.6 × 10^6^
	(9%)	(5%) ^1^	(14%) ^1^	(8%) ^1^
“−”	5.7 × 10^6^	4.0 × 10^6^	2.2 × 10^6^	3.0 × 10^6^
	(76%)	(74%)	(51%)	(64%)

^1^ For each biological replicate, percentage of aligned reads per total number of read.

**Table 2 microorganisms-10-01301-t002:** TSSs detected using dRNA-seq compared to previously mapped **TSS** with other methods.

Gene	Product	Label in MaGe ^1^	Promoter Region ^2^			5-URT (nt)	Approach ^3^	Comments
			−35	−10 + 1	/Name			
*dcmA*	DCM dehalogenase	METDI2656	TTGACA <16 nt> TATAGAactagccc **G**/P_A_ TTGACA <16 nt> TATAGAactagc **C**/P_A_ TTGACA <17 nt> TATAGTcaagtc **A**/P_A_			173 175 122	dRNA-seq nuclease S1 PE	*M. extorquens* DM4 (this study). TSS position 2562877 (plus) *M. extorquens* DM4 [16]. TSS identified from cloned DNA *Methylophilus* sp. DM11 [23]
*dcmR*	transcriptional regulator of DCM dehalogenase	METDI2655	TTGCGC <17 nt> TAACTAcaagg **G**/P_R1_ TTGCGC <17 nt> TAACTAcaagggtct **C**/P_R1_ TTTACT <16 nt> TTTACTcatcgg **A**/P_R2_ TTTACT <16 nt> TTTACTcatcgg **A**/P_R2_			66 62 157 157	dRNA-seq nuclease S1 dRNA-seq nuclease S1	*M. extorquens* DM4 (this study). TSS position 2562332 (minus) *M. extorquens* DM4 [16]. TSS identified from a cloned DNA *M. extorquens* DM4 (this study). TSS position 2562423 (minus) *M. extorquens* DM4 [16]. TSS identified from cloned DNA
*glyA*	serine hydroxyl-methyltransferase	METDI3959 META1_3384	TTGGCC <18 nt> ACGAATagtgc **C** ATCACC <16 nt> TGCCGCggcgtgta **C**			75 84	dRNA-seq PE	*M. extorquens* DM4 (this study). TSS position 3888433 (minus) *M. extorquens* AM1 [19]. Other TSS with 63 and 38 nt 5′UTR
*lspA* ^4^	lipoprotein signal peptidase	METDI3108 META1_2328 (old name ORF181)	TTCCCC <17 nt> TAGAAGcgctcca **A**/P*_ileS_* TCGACG <19 nt> GGTGCCcgagcggg **C**/P*_orf181_*			150 129	dRNA-seq PE	*M. extorquens* DM4 (this study). TSS position 3023525 (plus), 150 nt upstream of *ileS*. No TSS found upstream (0.9 kb) *lspA* *M. extorquens* AM1. P*_orf181_* deduced from a faint primer extension band, with low promoter activity (225 bp upstream of *orf181pqqFG* fused to *xylE* [18]
*mxaF*	methanol dehydrogenase alpha SU precursor	METDI5145 META1p4538 nd ^5^	AAGACA <18 nt> TAGAAAatatag **G** AAGACA <18 nt> TAGAAAatata **GG** AAGACA <18 nt> TAGAAAcgat **A**			168 167–168 170	dRNA-seq nuclease S1, run-off PE	*M. extorquens* DM4 (this study). TSS position 5068109 (minus) *M. extorquens* AM1 [15]. Gene *mxaF* also named *moxF* *M. organophilum* XX [21]
*mxaW*	uncharacterized conserved exported protein	METDI5146 META1p4539 nd	TTGACC <18 nt> ACCGTTgtcgtc**A**acggg**C** TTGGCA <nd> ACCCAT <nd> **G** ^6^ TTGACC <18 nt> ACCACTaggcgg **A**			41 52 54	dRNA-seq PE PE	*M. extorquens* DM4 (this study). TSS position 5068271 (plus) *M. extorquens* AM1 [18]. Gene *mxaW* also named *moxW* *M. organophilum* XX [22]
*mtkA*	malate thiokinase large SU	METDI2482 META1_1730	TTCCCG <17 nt> GAAGGTcggcccaa **C** TTGAGA <19 nt> AGTAATttttcc **G**/P*_qsc_*_2-1_ AAGTCA < ^7^ > AAGAAAaattga **G**/*P_qsc_*_2-2_			31 46 80	dRNA-seq PE	*M. extorquens* DM4 (this study). TSS position 2393413 (plus) *M. extorquens* AM1 of operon *qsc2* (*mtkA-mtkB-ppc-mcl*) [19]. Predominance of the two TSS varies upon growth conditions
*pqqA*	coenzyme PQQ biosynthesis protein A	METDI2503 META1_1751	TGGCGC <19 nt> TGATGGcgcc **A**/P*_mxbM_* TTGCAG <16 nt> CGATATacctccg **G**/P_pqqD_			119 95	dRNA-seq PE	*M. extorquens* DM4 (this study). TSS position 2416334 (minus) upstream of adjacent gene *mxbM* (METDI2504). No TSS detected upstream of METDI2503 *M. extorquens* AM1 [17]. Promoter checked using *xylE* fusion
*sga*	serine-glyoxylate aminotransferase	METDI2478 META1_1726	TTGCGC <16 nt> CGGGATcgccccc G/P*_sga_* GTGCCC <18 nt> CCGGCAgaggtg **C**/P*_qsc1_* TTGAAT <17 nt> CATCGAgggtt **C**/P*_qsc1_*			46 356 343	dRNA-seq PE	*M. extorquens* DM4 (this study). TSS position 2388788 (plus, P class) and 2388435 (plus, O class) *M. extorquens* AM1 of operon *qsc1* (*sga-hpr-mtdA-fch*) [19]

^1^ MaGe platform website: https://mage.genoscope.cns.fr/microscope/mage/index.php?; ^2^
**TSS**; ^3^ PE, primer extension; ^4^ *ileS* and *lspA* genes overlap over 3 nt in DM4 and AM1 strains suggesting a co-transcription.; ^5^ nd, data not available; ^6^ Sequence not found in AM1 sequenced genome [6], ^7^ Sequence of 30 nt was found between the −35 and −10 boxes in AM1 genome sequence [6].

## Data Availability

See Section 2.6: Data Deposition.

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
