# Peer review of "Genome-Wide Transcription Start Sites Mapping in Methylorubrum Grown with Dichloromethane and Methanol"

_microorganisms, 2022, doi:10.3390/microorganisms10071301_

Round 1

Reviewer 1 Report

This MS is intersting to readers. However, it need revision as listed in the following.

Introduction: There are many simple methods for TSS prediction and determination of specific target genes or gene clusters. Is genome-wide dRNA-Seq necessary? Please add more discussion on the significance of this study in

Table 2: There is layout error on the header of column 4.

Line 39-40Saying this study will help to find novel genes that are involved in methyl metabolism. What's the basis for this statement? Do you have data to support this statement.

Author Response

We thank Reviewer 1 for this speedy review. The manuscript has been revised according to the referees’ comments. All revisions made to the manuscript were marked up using the “Track Changes” function using MS Word/LaTeX. Detailed answers to the referees’ comments are provided in the following (italics). In addition, the English has been corrected.

Reviewer 1: This MS is interesting to readers. However, it needs revision as listed in the following.

Authors response: All the requested modifications have been taken into account.

Reviewer 1. Introduction: There are many simple methods for TSS prediction and determination of specific target genes or gene clusters. Is genome-wide dRNA-Seq necessary? Please add more discussion on the significance of this study in

Author response: The introduction was modified accordingly, with additional revision and reorganization to accommodate a newly added explanatory paragraph.

Reviewer 1. Table 2: There is layout error on the header of column 4.

Author response: Thank you for pointing this out, and corrected accordingly.

Reviewer 1. Line 39-40:Saying this study will help to find novel genes that are involved in methyl metabolism. What's the basis for this statement? Do you have data to support this statement.

Author response: Orphan TSS may indicate the expression of hitherto undetected genes. These novel genes may code small proteins or non protein-coding RNAs. All detected TSS in this study were determined from cultures obtained under methylotrophic growth conditions. Therefore, it is expected that at least a few of these genes will be associated with methylotrophic metabolism.

Reviewer 2 Report

The manuscript is related to mapping of the TSS of the genome of M. extorquens DM4 grown under methylotrophic mid-exponential phase cultures by using of dRNA89 seq. Effect of dichloromethane (DCM) was compared to the effect of methanol. The obtained qualitative transcription details can be used in future studies in which transcriptional regulation is crucial for one-carbon compound utilization and efficient DCM removal at polluted sites. The topic is very important from ecological point of view and the study deserve to be published.

The introduction is well written and enough informative. The material and methods are clear. The discussion properly reflects the obtained data. The references are on the topic and precisely selected.

Few minor editorial corrections are necessary:

Page 1, Line 23: the word “unprecedented” can be deleted, the sentence is very redundant.

Page 3, Line 102: “Biological replicates were performed for each of the two tested conditions.” – how many replicates were analysed?

Author Response

Author response to the Reviewer 2 comments 

Authors: We thank the reviewers for this speedy review. The manuscript has been revised according to the referees’ comments. All revisions made to the manuscript were marked up using the “Track Changes” function using MS Word/LaTeX. Detailed answers to the referees’ comments are provided in the following (italics). In addition, the English has been corrected.

Reviewer 2: The manuscript is related to mapping of the TSS of the genome of M. extorquens DM4 grown under methylotrophic mid-exponential phase cultures by using of dRNA89 seq. Effect of dichloromethane (DCM) was compared to the effect of methanol. The obtained qualitative transcription details can be used in future studies in which transcriptional regulation is crucial for one-carbon compound utilization and efficient DCM removal at polluted sites. The topic is very important from ecological point of view and the study deserves to be published.

The introduction is well written and enough informative. The material and methods are clear. The discussion properly reflects the obtained data. The references are on the topic and precisely selected.

Author response: Thank you for these comments.

Reviewer 2: Few minor editorial corrections are necessary: Page 1, Line 23: the word “unprecedented” can be deleted, the sentence is very redundant.

Author response: Done. In the last sentence of the paragraph,” provides unprecedented qualitative transcription details that» was deleted.

Reviewer 2. Page 3, Line 102: “Biological replicates were performed for each of the two tested conditions.” – how many replicates were analysed?

Author response: The sentence was changed from « Biological replicates were performed for each of the two tested conditions » to « Two biological replicates were performed for each of the two tested conditions”.